# Neural Network Explainable AI Based on Paraconsistent Analysis: An Extension

Francisco S. Marcondes *[ID], Dalila Durães *[ID], Flávio Santos [ID], José João Almeida [ID] and Paulo Novais [ID]

ALGORITMI Centre, University of Minho, 4710-057 Braga, Portugal; flavio.santos@algoritmi.uminho.pt (F.S.); jj@di.uminho.pt (J.J.A.); pjon@di.uminho.pt (P.N.)
* Correspondence: francisco.marcondes@algoritmi.uminho.pt (F.S.M.); dalila.duraes@algoritmi.uminho.pt (D.D.)

**Abstract:** This paper explores the use of paraconsistent analysis for assessing neural networks from an explainable AI perspective. This is an early exploration paper aiming to understand whether paraconsistent analysis can be applied for understanding neural networks and whether it is worth further develop the subject in future research. The answers to these two questions are affirmative. Paraconsistent analysis provides insightful prediction visualisation through a mature formal framework that provides proper support for reasoning. The significant potential envisioned is the that paraconsistent analysis will be used for guiding neural network development projects, despite the performance issues. This paper provides two explorations. The first was a baseline experiment based on MNIST for establishing the link between paraconsistency and neural networks. The second experiment aimed to detect violence in audio files to verify whether the paraconsistent framework scales to industry level problems. The conclusion shown by this early assessment is that further research on this subject is worthful, and may eventually result in a significant contribution to the field.

**Keywords:** paraconsistent logic; explainable AI; neural network





## 1. Introduction

In the last decade, the success of artificial intelligence (AI) applications, namely, applications that use machine learning (ML) and/or deep learning (DL) models, has been resounding, as they offer broad benefits and are applied in several areas. However, these applications are not able to logically explain their autonomous decisions and actions to human users. Although explanations may not be essential for specific applications, for many critical applications, such as agriculture and environmental projects [1–4], traffic-flow management and object detection [5,6], and ailment cues [7], explanations are essential for users to understand, trust, and effectively manage these new artificially intelligent partners [8].

A method used to explain machine learning or deep learning models' outputs is called explainable AI (XAI) [9]. The interest in XAI is rising, as AI is beginning to be used in increasingly sensitive environments where safety and privacy must be assured. The motivation for XAI is that neural networks are black boxes, and there is no guarantee that a model uses sound reasoning when evaluating an input. In other words, the features and relations established by such a model are not accessible, and usually are very different from what would be expected from a human perspective, which leads to unpredictability in how a model responds to certain situations. Thus, it is necessary to use a system based on a human perspective to explain how it works.

Paraconsistent logic was used embedded in neural networks for enhancing the training procedure [10–12]. Since the objective is not to trace the training process but trying to make sense about the generated outputs, the approach of embedding paraconsistency in the neural model was not pursued. Paraconsistent analysis was then used for assessing the the model's output as an external approach.

The idea of using paraconsistent logic for assessing neural networks outcomes emerged from the outcome evaluation of an industry level model. It was noticed that the outcomes had an odd shape: the model was able to identify one class but became random with another. Therefore, the precision value was masked. For an instance, consider a balanced dataset. Let the model accurately identify 50% of the test entries and randomly classify the 50%. The result should be greater than 75% accuracy, though mistakenly identifying different entries on each run. On a such model, the actual accuracy should not be considered 75% but 50%.

After further studying the issue, it was realised that the classes used were a class and its negation. This led to an explosion situation (the negation class is the compliment, and therefore an open set). Since paraconsistent logic was developed for handling explosions, it was natural candidate to be explored. To the best of the author's knowledge, paraconsistency was never used as an analysis tool applied for explainable AI (XAI). Therefore, this article's objective is to report the findings of an initial attempt at using paraconsistent analysis as an explainable approach. Our purpose was assess whether further research on the topic should be stimulated.

Paraconsistent analysis handles explosions by providing a bi-dimensional data visualisation by including the axes of inconsistency and para-completeness. Therefore, it enables, in addition to the classification, one to verify whether the identified features belong to both or neither classes. It is helpful for analysing anomalous behaviour, including positive and false negatives. Therefore, proven useful, paraconsistent analysis may be a cornerstone for neural models' design and assessment.

*Organisation*

Being an extended version of [13], the aim is to present a lengthier exposition of the ideas presented on the base paper. The research objective was to evaluate the use of annotated paraconsistent analysis for assessing neural networks' outputs. The research is currently at an early stage whereat explorations are ongoing and feasibility is being evaluated.

The remainder of this paper is structured as follows: In Section 2 are the main works related to the annotated paraconsistent logic. Section 3 presents an analysis of the annotated paraconsistent logic with the fundamental principles. In Section 4, we describe the results of the MNIST dataset analysis. In addition, we also present in Section 4 the audio-based violence detection. Section 5 presents the discussions about the results obtained in the previous section. The concluding remarks and future works are given in Section 6.

## 2. Related Works

Paraconsistent logic is especially suited for tackling *contradiction* [14]. With a contradictory model it is possible to prove true both a statement and its *negation* ($p \wedge \neg p$), called *explosion*. This is related to the fact that $\neg p$ is the *complement* of $p$, and therefore, an *open set* [15]. Paraconsistent logic handles this issue by assessing the element's favourable and contrary pieces of evidence, balancing them within a lattice. In this paper, context, $V$ (true) and $F$ (false) are classes of a binary model, and the evidence is a model prediction for each class. According to the position of an element in the lattice, it is possible to understand its fitness to a class or its indeterminacy.

Currently, most classification efforts are based on closed sets—i.e., all classes are known a priori. When an alien class is to be recognised, the result is likely to be random, a serious safety issue, as these models are being built aiming at open world situations. Therefore, the explosion and the open set issues in neural networks must be addressed [16]. Uncertainty and unknown are different concepts, yet paraconsistent analysis is capable of addressing both [17]. It assesses if an input relates to a class (true) or another (false), and two types of unknown, both classes ($\top$) or neither classes ($\bot$); it also provides a continuous uncertainty degree connecting the lattice extremities [14].

The open world recognition problem is intrinsically problematic due to the infinity property of open sets. Some attempts at handling this problem range between early approaches, such as the use of a threshold [18] and a "garbage" (or "background") class [19], to more recent ones, such as the open-max, which uses the extreme value theory within a neural network for calibrating the compact abating probability [20], and an approach that identifies unknown classes through a proposed loss function [21]. To the best of the author's knowledge, this is the first effort of using annotated paraconsistent analysis as an XAI alternative for addressing the open set issue in neural networks.

## 3. Annotated Paraconsistent Logic and Analysis

The law of excluded middle $p \vee \neg p$ is based on the principle that an assertion is true or false. Therefore, the principle of explosion states that anything follows from $p \wedge \neg p$ within syllogistic reasoning. As presented by the intuitionism, see [22], the excluded middle does not stand in the real world, as partially true statements exist. For tackling situations as such, paraconsistent logic was developed. In short, a statement may be true or false ($p \vee \neg p$); true and false ($p \wedge \neg p$); and neither true nor false ($\neg p \vee \neg \neg p$) to different degrees. From a set theory perspective, a set is closed *iff* its complement is open [15]; therefore, the explosion principle can be restated as $P \vee \bar{P}$. In other words, the explosion on the $\neg p$ is a consequence of the open set $\bar{P}$ infinity. For this paper's purposes, $T$ and $F$ are considered classes; therefore, only binary classification is considered. In addition, the favourable and contrary evidence ($\lambda$ and $\mu$) are considered the output of the network model. In this sense, `sigmoid` is preferred over `softmax` as the activation function for keeping the probabilities independent and avoid loosing information [13]. Figure 1 summarises the main constraints of paraconsistent logic.

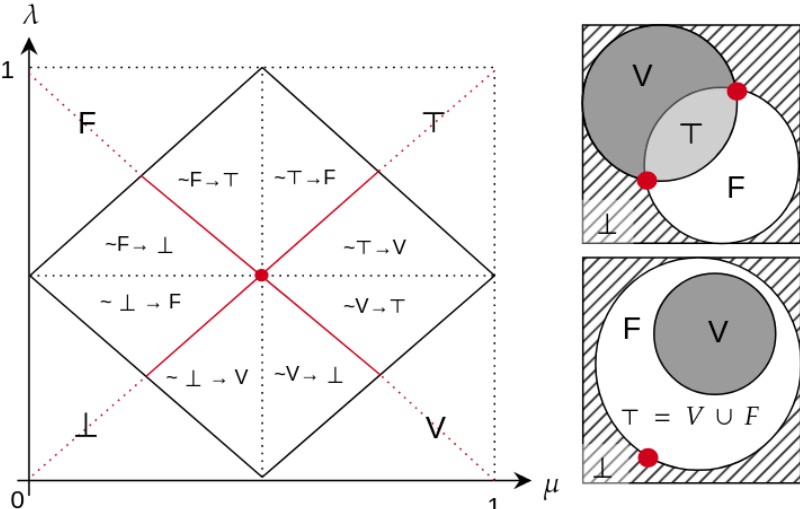

**Figure 1.** a Paraconsistent lattice (with analogous set representations; see [13]): $V$ = true $(0, 1)$; $F$ = false $(1, 0)$; $\top$ = inconsistency $(1, 1)$; $\bot$ = para-completeness $(0, 0)$; and the red dot = indefiniteness $(0.5, 0.5)$ [14]. The line $[(1, 0), (0, 1)]$ is probability and the line $[(0, 0), (1, 1)]$ is pertinence. $\lambda$ and $\mu$ are the degrees of favourable and contrary evidence. The "$\sim$" symbol means "almost" and $\rightarrow$ means "tending"; e.g., the expression $F \rightarrow \top$ means almost-false tending to inconsistency.

The paraconsistent analysis is an approach often used for supporting decisions (embedded in a system or not) [14]. On the other hand, this paper aims to use paraconsistent logic for analysing the output of neural networks to understand how the model is behaving, and evaluates its strengths and weaknesses. Two types of analyses were undertaken. The first was to plot the prediction for the test set for obtaining a performance overview, and the second was to reduce (since this is an initial paper, and the reduction was performed by computing the averages of the resulting predictions, in future works, potentially

better alternatives will be assessed) that performance into a single point that summarises the behaviour and provides the degree of certainty achieved.

## 4. Results

### *4.1. MNIST Dataset Analysis*

In this section, the discussion is based on the MNIST dataset; see [23] from the paraconsitent; see the perspective of [14]. It discusses the differences between `softmax` and `sigmoid` activation functions suggesting that `softmax` loses paraconsitent information that is kept by the `sigmoid` [13]. In addition, it exemplifies the problem of open space faced by neural networks.

#### 4.1.1. Dataset and Model Description

The Modified National Institute of Standards and Technology (MNIST) is a dataset composed of seventy thousand, $28 \times 28$ pixel, grey-scale, labelled, handwritten numbers [24]. As a high quality, widespread and extensively studied dataset, it is also considered the "Hello World!" of deep learning [23]. As this paper is concerned with binary classification, the MNIST is reduced to two classes for proper support. The selected classes are "zero" and "one", resulting in a balanced reduced dataset with 14,780 samples (12665 for training and 2115 for testing). A simple two layer sequential model retrieves the expected accuracy of $\approx$99% after five epochs, for both the reduced MNIST training and test sets. For reference, the resulting confusion matrix for the test set using `softmax` is $\begin{bmatrix} 979 & 0 \\ 1 & 1135 \end{bmatrix}$. For reproduction, the seed used in training is "1234567."

#### 4.1.2. Analysis of the Network

The first issue to consider is whether the network properly managed to classify the samples, or, as the case study presented in [13], was the result of an explosion (the model was able to identify one class but randomised the classification for another class). As depicted by Figure 2a, the classes are plainly detached, suggesting that the model was able to classify the provided instances effectively. This same plot but shaped into the paraconsistent lattice is presented in Figure 2b.

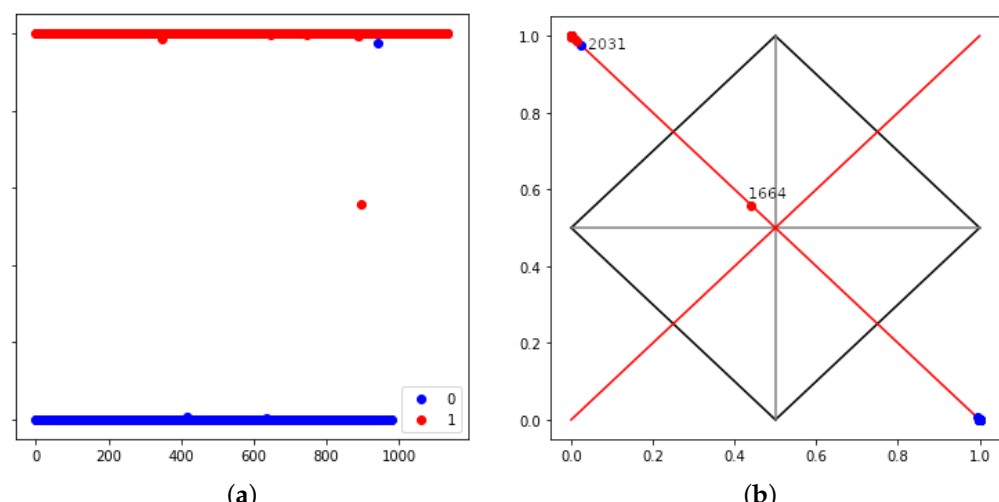

**Figure 2.** Prediction plot for the model trained with `softmax` activation. The blue is used for referring to images labelled with zero, and the red to images labelled with one.

Notice that in Figure 2b all points overlap upon the probability line. This is because the model was trained with the `softmax` activation that reduces the output into probabilities where the sum is always one. For reference, the outlier points are labelled in the plot and presented in Figure 3. Notice that point 1664, despite being correctly classified, presents a high degree of uncertainty, and point 2031 is mistaken even with high probability. However,

as `softmax` narrows the analysis into the probability line, the other dimensions provided by the paraconsistent lattice are lost [13].

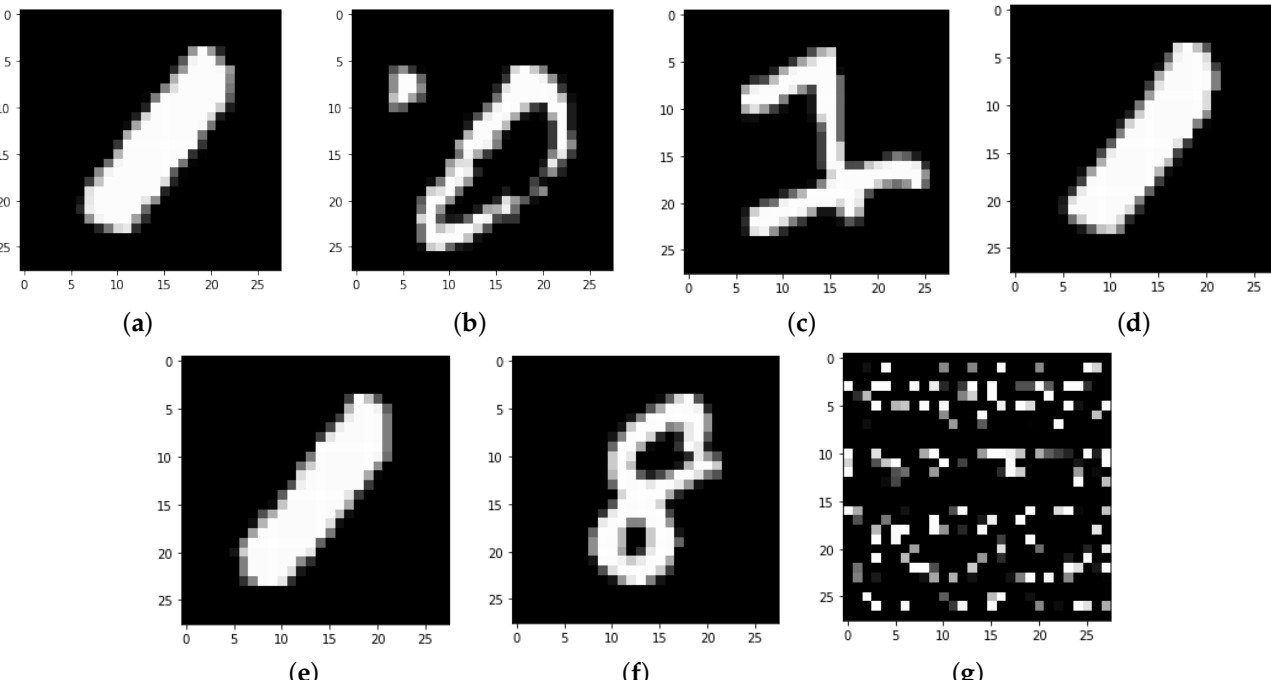

**Figure 3.** Presentation of some elements of interest from the sample: (**a**) point 1664 depicted in Figures 2b and 4; (**b**) point 2031 depicted in Figures 2b and 4; (**c**) point 626 depicted in Figure 4; (**d**) point 1643 depicted in Figure 4; (**e**) point 1645 depicted in Figure 4; (**f**) an instance for the number eight used as input for the prediction model (see Figure 5a); and (**g**) an instance for the random generated input used as input for the prediction model (see Figure 5b).

After the replacing the `softmax` by the `sigmoid` the result depicted in Figure 4 becomes more informative. Point 2031 (Figure 3a) being to the right of the probability line means that it possesses features common to both classes. The same for the point 1664 (Figure 3b). Both points are on similar distances from their labelled classes yet within the uncertainty square. On the other hand, point 626 (Figure 3c) is in the para-completeness quadrant, i.e., its features are not likely to be neither of one nor of zero.

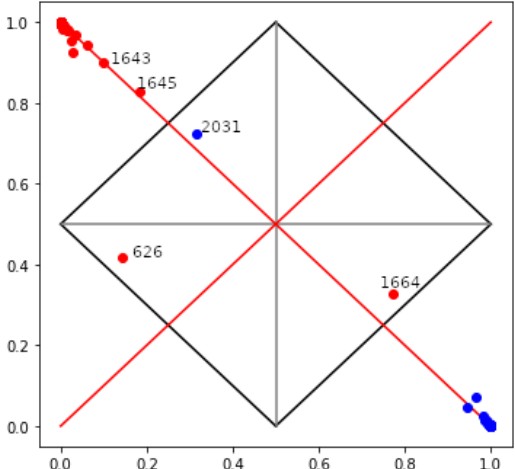

**Figure 4.** Prediction plot for the model trained with `sigmoid` activation.

The points 1643 and 1645 (Figure 3d) are interesting, since resembles the point 1664, yet classified with a major interval. In order to explore this issue, from the MNIST dataset was

extracted, only the images were labelled as eight and submitted to the model (see Figure 3f for reference). The result is an explosion as presented in Figure 5a with 40.7% of the sample being classified as zero and the other 59.3% as one. Something analogous happens when shuffling the image pixels for producing a synthetic dataset with meaningless images (refer to Figure 3g). The classification results in 57.6% for zero and 42.4% for one, the scatter is presented in Figure 5b.

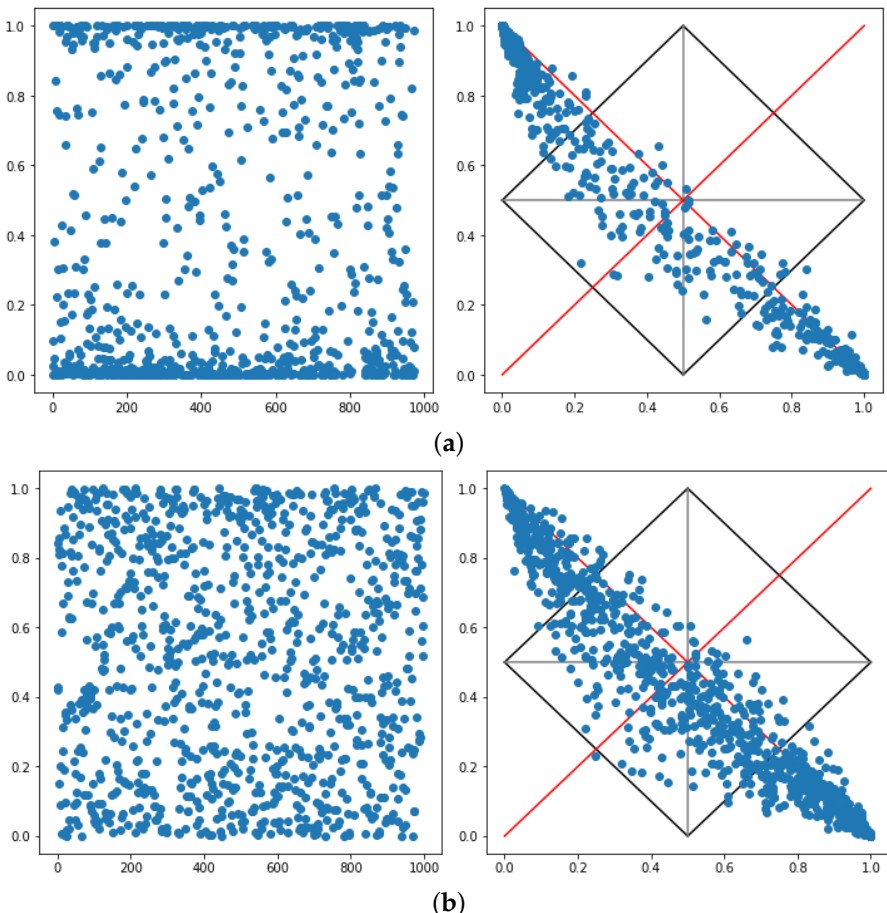

**Figure 5.** Plot on predictions of number eight (**a**) and randomly generated images (**b**) for a model trained to classify zero or one.

This result is entirely counter intuitive since the expected concentration would be in the centre of the diagram and scattered around the pertinence line. The issue that arises is that this machine learning models lose themselves when facing an unexpected input. This is a prohibitive outcome when considering projects expected to interact with the world of the problem, therefore another model should be pursued. Notice that this is not a problem of false positive or false negative but of an inconsistent input.

After computing the average of values, the results come to be more like that would be the expected (refer to Figure 6). Remark that average is perhaps not the best approach for such reduction as, based only on it, it is not possible to know if the average point refers to a bunch of points on a neighbour region or a scattered shape as it is.

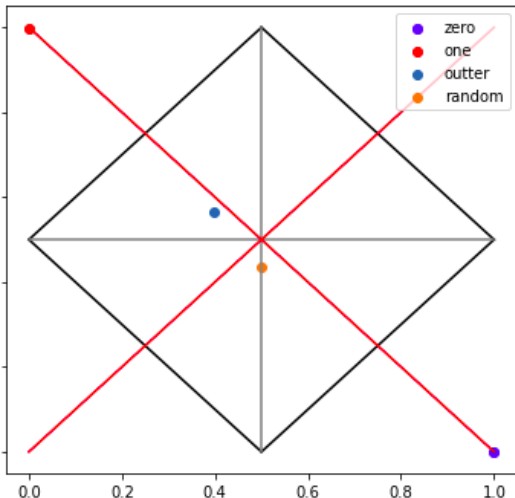

**Figure 6.** Average values of outputs from Figure 4 (zero and one classes) and Figure 5a (outer and random classes).

Nevertheless, it is interesting that the random point falls precisely in the boundary of two quadrants. Situations like that are expected to be as uncommon, as the indefiniteness point (0.5, 0.5). Therefore, extrapolating this result may require stating that the lines [(0, 0.5), (1, 0.5)] and [(0.5, 0), (0.5, 1)] are lines of indefiniteness. Therefore, the random point is indefinite with high certainty due to its proximity to the indefiniteness point. In addition, it also makes sense that the outer point holds on the $\sim 1 \rightarrow \perp$ region, since it has features that are neither zero nor one but with high indefiniteness.

### 4.2. Audio-Based Violence Detection

In this section, the discussion is an extension of the one presented in [13] targeting the training of a model for identifying violence from audio. This instance is interesting as audio presumably mixes several features yielding to both inconsistent and para-complete states. Different from the previous experiment that is a classical (i.e., extensively addressed and discussed) instance, this is closer to a typical industry situation.

#### 4.2.1. Dataset and Model Description

We wanted to have a dataset with violent and non-violent audio scenes for this experiment. However, it was impossible to find a suitable dataset because audio did not exist in all scenes in some datasets. To solve this situation, we created a new dataset based on the audio RLVS dataset [25] (existing in some videos) and Uber audio scenes from YouTube. As explained in [25], the RLVS dataset is a binary class dataset composed of 2000 videos divided into violence and non-violence. The RLVS dataset extracted the audio channel of all videos, resulting in 193 non-audio files and 745 audio files. Then it was included complementary videos with audio following the same good resolution clips (480p–720p), variety in gender, race, age and several indoor and outdoor environments as defined by the RLVS [25]. The additional videos are also all English speech real-life videos collected from YouTube trimmed to three to seven seconds of duration. This derived dataset was then named RLVSwA (Real Life Violence Situations with Audio). This RLVSwA dataset has 517 clips, among which 176 show violent situations, and 341 do not show violence. Our RLVSwA dataset is unbalanced. However, we believe that is closer to reality, which has more non-violent situations than violent.

Instead of beginning with custom models, we have trained well-known models in the literature to find the better one for our task. We have performed experiments with two architectures, and they are CNN6 [26], and VGG19 [26].

### 4.2.2. Analysis of the Network

As in the previous experiment, the first issue to consider is if the network properly managed to classify the sample or resulted in an explosion. As depicted in Figure 7, both models managed to recognise *violence* but turned into random attribution when recognising *non-violence*. In other words, these networks' accuracy was due to *violence* detection plus a stochastic classification of *non-violence* samples.

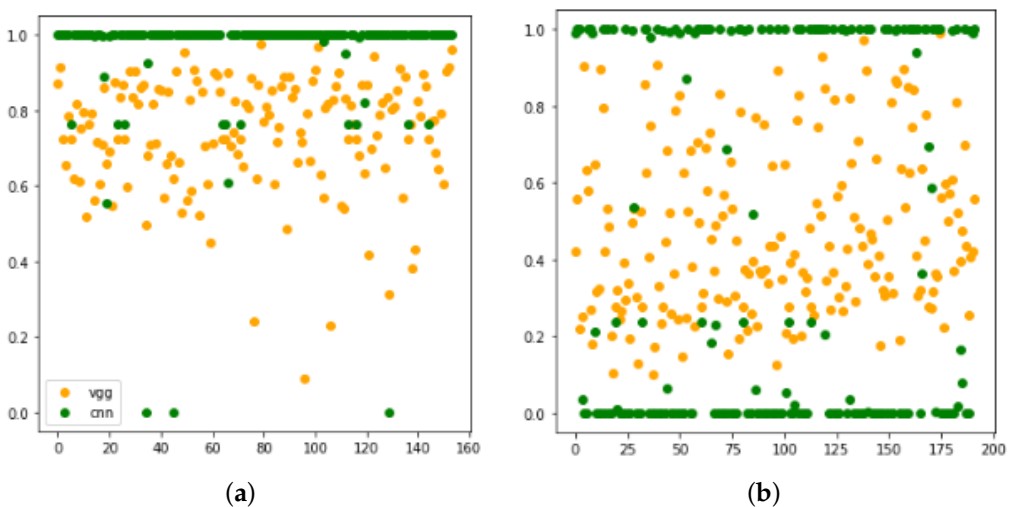

(a)　　　　　　　　(b)

**Figure 7.** Prediction plot for the VGG19 and CCN6 models trained with `softmax` activation: (**a**) test set labelled with "violence"; (**b**) test set labelled with "non-violence".

Notice that, despite *non-violence* being explicitly trained, the plot in Figure 7b is analogous to those presented in Figure 5 for non-trained classes. This is probably due to a search for the negation of violent features resulting in the explosion. The negation of a feature set means a search for the complement set; that in turn is an open set [15].

For this model the `softmax` was replaced by a `linear` activation, and the result set normalised through the $l_\infty norm$ using the `sklearn`. This is an improvement over the base paper, as the max-min scaler dragged the scattered points distorting the visualisation (refer to [13] for comparison). The result is plot in Figure 8.

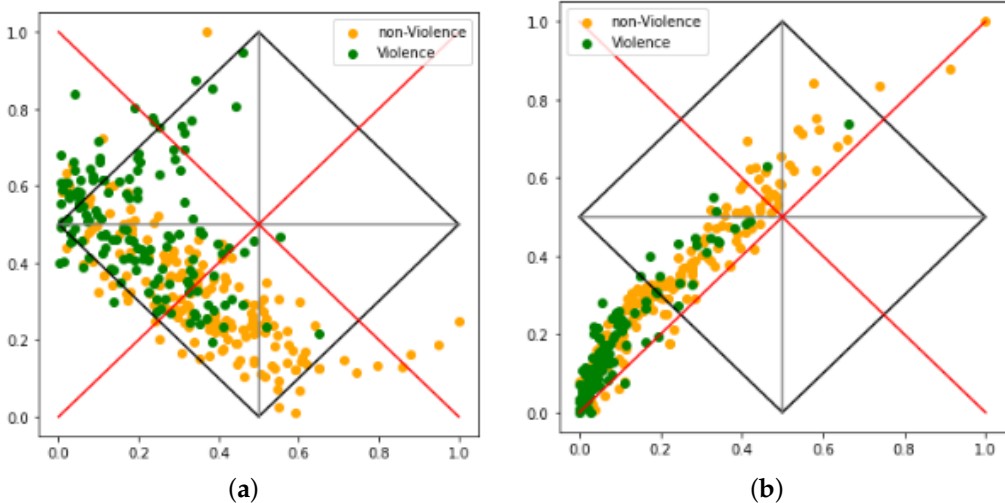

(a)　　　　　　　　(b)

**Figure 8.** Prediction plot for: (**a**) VGG; (**b**) CNN.

The shape of plots presented in Figure 8 are insightful in the sense that both models fall on para-completeness regions with VGG scattering through the probability (Figure 8a) line and CNN through the pertinence line (Figure 8b). In addition, the VGG model presents

a medium to high indefiniteness, suggesting that several features in the test set are neither from a class nor another, but, perhaps a subset of the many are (considering open world audio, such assertion makes sense). On the other hand, the CNN model presents low to medium indefiniteness but high para-completeness, suggesting the model could not devise a suitable set of features properly. In short, Figure 8 suggests that VGG is suitable for refinement, but CNN did not perform for that dataset, and it may not make sense to refine it further.

Finally, Figure 9 presents the average valued for the predictions presented in Figure 8. VGG (all labels) points fall in the $\sim\!\perp\,\rightarrow V$ region and CNN (all labels) in the $\perp$ region. VGG classes tend to fall in the correct places, but the same is not true for CNN.

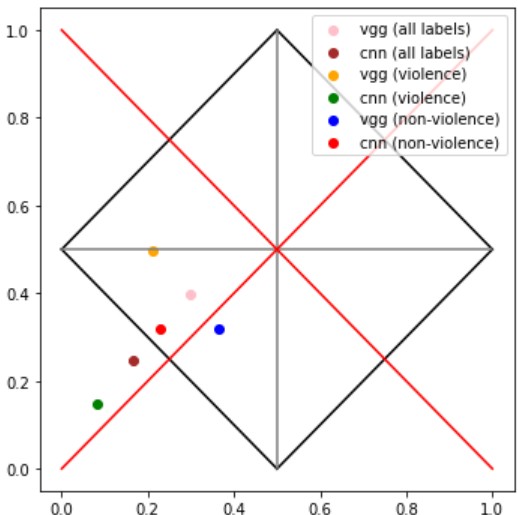

**Figure 9.** Average prediction values from Figure 8.

## 5. Discussion

Investigations of neural networks is widespread. This paper proposes an early attempt of using paraconsistent analysis for understating what types of contribution such an approach may deliver. Therefore, it is an exploratory paper for assessing the feasibility of using paraconsistency for evaluating neural networks' trustworthiness based on their output. The question to be answered is then whether such path should be further explored or not. The conclusion drawn in this section is: yes, it should.

Evaluating the explorations in this paper, paraconsistent analysis delivered insightful views that can be used both for understanding and for refining a particular model. An example of this is presented in Figure 5. It shows the model is predicting several of out-of-distribution inputs *with high confidence*: a counter-intuitive and unexpected behaviour. Consider an open-world environment and ignore that the model could be a safety threat.

From an XAI perspective, it also helps with evaluating and choosing between architectures. This can be realised on the plots in Figure 8. The VGG model scattered through the probability line, suggesting that the model managed to find the proper features for classification. The CNN model, on another hand, scattered around the pertinence line, suggesting that the model was not capable of finding suitable features. In such a scenario, it makes sense to concentrate efforts in improving the VGG model and abandon the CNN. Highlight that it is a different decision making process compared to evaluating metrics.

Considering specifically Figure 8b, it is interesting to notice that the distribution is concentrated in the para-completeness region. This suggests that, for most of audios, it was not possible to find features related to violence, nor to non-violence. However, there are also some points in the inconsistency region suggesting that features of both classes were found. Instances of this behaviour can be explicated by friendly yelling or rough music playing in the background. These situations are not found in Figure 8a; this suggests

that the chosen way to build the lattice is not yet well suited, requiring improvements in other research.

Another difficult issue to handle in neural networks is dataset bias. Bias is not easily identified, as its cause is often concealed in the dataset and remains hidden after the training process due to the model's opacity. The paraconsistent lattice provides a visualisation that may aid on detecting bias. Bias relates to inconsistency; therefore, there is a possibility to explore it if the paraconsistent lattice provides a proper visualisation for identifying biased datasets (based on the output). The expectation is to find a concentration in the inconsistent region of the lattice.

In the base paper it was suggested that the scattering shape could be related to performance. This is because the accuracy of the VGG network is 80% and that of the CNN, 60%; as already mentioned, VGG points scatter through the probability line and CNN through the pertinence line. Nevertheless, considering the plots in Figure 5, this could not be true. Therefore, paraconsistent analysis visualisation, presumably, is not related to its performance.

Another finding is that the paraconsistent threshold shown itself is not accurate for not-unknown class situations. In Figure 4, it properly contained the wrong answers, and in Figure 8a it properly contained the explosion. Notice that in Figure 8a the accuracy is 80%, but with a low paraconsistent confidence, yet with high adequacy as it behaved as expected given the explosion. MNIST, in turn, had 99% accuracy with high paraconsistent confidence and confidence. However, within an open set situation, the MNIST had low paraconsistent confidence and adequacy (the expected shape would be on the other line). In short, confidence lowers as it approaches the centre and adequacy, as the model behaves as should be expected. These parameters aid on decision support about the model's adequacy.

In summary, as paraconsistent analysis enables assessing the quality of a model, despite its performance, it may have the potential to be a guiding metric for neural network design and refinement. It could be used for directing strategies and models to adopt during elaboration. From a project perspective, especially on industry, it is worth assessing.

In addition, following the theory, binary classifications such as classifying an action as *violent* ($v$ or $V$) or *not violent* ($\neg v$ or $\bar{V}$) results on the explosion of the proposition negation (the complement set). The explosion caused by the negation is analogous to the prediction of a class that the model is not trained to identify (the open world issue). This second case, using this paper instance, could be expressed as $(0 \vee 1) \wedge \neg(0 \vee 1)$; therefore, both issues present the same shape. This results in a good practice proposal to be considered when modelling neural networks: *avoid the negation of a class*. This shows the importance of using a formal framework for investigating neural networks compared to ad hoc approaches.

Once again, this paper presents an early exploration; therefore, most of the presented results are initial and eventually incipient. Therefore, most of claimed deductions are yet conjectures and speculations about potential. Nevertheless, as the presented claims are plausible and may significantly improve the understanding, design and projections of neural networks, this theme should be further explored and deeply analysed in order to actually understand the contributions that it may offer.

## 6. Conclusions

This is an early paper on paraconsistent analysis used for assessing the predictions of neural networks. It presented an academic exploration based on MNIST and an industry related exploration for audio-based violence detection. The explorations suggest that paraconsistent analysis, specifically the paraconsistent lattice, provides insightful visualisation for predictions, being a potential explainable AI tool. Perhaps the most evident strength is to provide proper visualisation for issues related to explosion and open set situations, and another strength is proper handling of uncertainty and unknowns. Nevertheless, the good practice of avoiding negation classes is also an useful contribution.

A first finding, already discussed in the base paper [13], is that the `softmax`, and any function whose prediction summation is always one, is unsuited to paraconsistent analysis.

The claim is that `softmax` looses paraconsistent information, `sigmoid` being a preferred option. Nevertheless, a current drawback of paraconsistent analysis is that is capable of assessing only binary classifications.

Another finding is the paraconsistency confidence (approach or depart from the central point) and adequacy (suits with the expected behaviour) properties may be useful for supporting the utility of the model, given its application context. Additionally, finding a properly way to reduce the scattering to a single point may be a proper way to understand the prediction result and the behaviour of the network. Nevertheless, as it has its own limitations, a composed presentation with both plots may be preferred.

The presented results support the idea that the class negation (the open set) may be a problematic issue in neural networks. Although this is a known issue, paraconsistentcy managed to present a different perspective of the problem. It also provided a visual way of comparing the performances of two distinct networks used for solving a same problem. It showed, crystal clear, *why* the VGG is a better choice for this problem than the CNN. Something similar was found for dataset bias.

Perhaps the most important prospect raised for the proposed approach is to be used for guiding neural network projects by assessing their quality, despite its performance. Being a mature formal framework, it enables one to assess the resulting lattice from several perspectives, delivering insightful information, as presented during the paper.

The major limitation of the current approach is being applicable only to binary models. A multi-class network would require a multi-dimensional lattice. For future works, it is suggested to deepen the explorations and feasibility for better assessing the proposed approach. The first issue to address is to find the best suited parameters for properly building the lattice; after that, one ought perform a systematic evaluation upon different types of model. Accordingly to these results, it will be possible to fully understand the outcomes of a paraconsistent analysis and its actual utility for neural network models.

**Author Contributions:** Conceptualization, F.S.M., F.S. and D.D.; Supervision, J.J.A. and P.N. All authors have read and agreed to the published version of the manuscript.

**Funding:** This work is financed by National Funds through the Portuguese funding agency, FCT—Fundação para a Ciência e a Tecnologia within project DSAIPA/AI/0099/2019.

**Conflicts of Interest:** The authors declare no conflict of interest.

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
