# Peer review of "Neural Network Explainable AI Based on Paraconsistent Analysis: An Extension"

_electronics, doi:10.3390/electronics10212660_

Round 1

Reviewer 1 Report

The research is aimed at evaluating neural networks using the method of paraconsistent analysis from an explainable AI perspective. The paper is organized in 5 sections. They present an analysis of the paraconsistent logic with some basic principles. Authors describe the results of the MNIST dataset analysis and audio-based violence detection.

I would recommend the authors to make the following methodological changes in their paper, which I believe would improve the presentation of the study:

  • to extend the introduction by justifying the importance of the research methodology used and its application in neural networks;
  • to clearly formulate the purpose and objectives of the paper, including in the introduction;
  • to expand the sources of the study to be considered in the "Related Works" section;
  • to improve the conclusion by synthesizing the findings of the authors. 

Author Response

Comment 1: extend the introduction by justifying the importance of the research methodology used and its application in neural networks.
Reply: We expanded the introduction with three paragraphs trying to clarify these issues.

Comment 2: clearly formulate the purpose and objectives of the paper, including in the introduction.
Reply: We add a clear statement of the paper objective and purpose in the introduction.

Comment 3: expand the sources of the study to be considered in the Related Works section
Reply: This is precisely a problem that this paper tackle, we were not able to find many related works; those that we found are included in the paper.  

Comment 4: improve the conclusion by synthesizing the findings
Reply: We improved the conclusion for synthesizing the findings

Reviewer 2 Report

1) Page 1, line 21 Add an application of agriculture and cite the following papers

A) A. Farooq, X. Jia, J. Hu and J. Zhou, "Transferable Convolutional Neural Network for Weed Mapping With Multisensor Imagery," in IEEE Transactions on Geoscience and Remote Sensing, doi: 10.1109/TGRS.2021.3102243.

B) A. Farooq, J. Hu and X. Jia, "Weed Classification in Hyperspectral Remote Sensing Images Via Deep Convolutional Neural Network," IGARSS 2018 - 2018 IEEE International Geoscience and Remote Sensing Symposium, 2018, pp. 3816-3819, doi: 10.1109/IGARSS.2018.8518541.

also add the references for the other applications
2) Page 1, line 34: Instead of “This paper“ best will be “ In this article,” 
3) Page 1, line 34: Explain the method in details why the authors didn’t follow the direction.

4) Page 3, Line 99: dataset cf what is cf?

5) Page 3, Line 100: why softmax and sigmoid is with text?

6) Figure 2 Need explanation in the label of figure a and b.

7) From where these points 2031, 1664 comes from. What Is the relation with the softmax and sigmoid.

8) Hard to understand the explanation of Figure 3.

Author Response

Comment 1: Page 1, line 21 Add an application of agriculture and add the references for the other applications
Reply: We reviewed the references as suggested.

Comment 2: Instead of "This paper" best will be "In this article"
Reply: We had adjusted the text.

Comment 3: Explain the method details why the authors didn't follow the direction
Reply: We enhanced the explanation in order to further clarify this issue.

Comment 4: what is cf?
Reply: It is the abbreviation for the Latin word confer/conferatur. It is used to refer the reader to other material. Nevertheless, we change all occurrences to 'see'.

Comment 5: why softmax and sigmoid with text?
Reply: MNIST is an image data-set.

Comment 6: Need explanation in the label of figure 2 a and b
Reply: We further explained the labels in the figure caption.

Comment 7a: From where these points 2031 and 1664 comes from.
Reply: From the MNIST dataset.

Comment 7b: What is the relation between the softmax and sigmoid?
Reply: I did not understand if this is a general question about the overall relation between softmax and sigmoid or a specific one, such as their relationships in the context of Figure 2.

Comment 8: Hard to understand the explanation of Figure 3.
Reply: We enhanced the figure caption explanation.
